# Traceable Scheme of Public Key Encryption with Equality Test

**DOI:** 10.3390/e24030309

**Published:** 2022-02-22

**Authors:** Huijun Zhu, Qingji Xue, Tianfeng Li, Dong Xie

**Affiliations:** 1School of Digital Media and Art Design, Nanyang Institute of Technology, Nanyang 473004, China; xue_qj@sina.com (Q.X.); 3071066@nyist.edu.cn (T.L.); 2State Key Laboratory of Networking and Switching Technology, Beijing University of Posts and Telecommunications, Beijing 100876, China; 3Graphic Image and Intelligent Processing in Henan Province, International Joint Laboratory, Nanyang Institute of Technology, Nanyang 473000, China; 4School of Computer and Information, Anhui Normal University, Wuhu 241002, China; xiedong@ahnu.edu.cn

**Keywords:** public key encryption, equality test, blockchain, cloud server

## Abstract

Public key encryption supporting equality test (PKEwET) schemes, because of their special function, have good applications in many fields, such as in cloud computing services, blockchain, and the Internet of Things. The original PKEwET has no authorization function. Subsequently, many PKEwET schemes have been proposed with the ability to perform authorization against various application scenarios. However, these schemes are incapable of traceability to the ciphertexts. In this paper, the ability of tracing to the ciphertexts is introduced into a PKEwET scheme. For the ciphertexts, the presented scheme supports not only the equality test, but also has the function of traceability. Meanwhile, the security of the proposed scheme is revealed by a game between an adversary and a simulator, and it achieves a desirable level of security. Depending on the attacker’s privileges, it can resist OW-CCA security against an adversary with a trapdoor, and can resist IND-CCA security against an adversary without a trapdoor. Finally, the performance of the presented scheme is discussed.

## 1. Introduction

With the continuous development of the Internet of Things (IoT), the security of data has gotten more attention. In order to ensure the security of data, data are stored on a server by encryption. However, it is inconvenient for effective application when the data are encrypted, making it impossible to search within encrypted data. Therefore, searchable encryption (SE) is presented [1]. The aim of SE is to produce a tag related to ciphertext, and to classify the ciphertexts. Since this primitive approach was proposed, many cryptographers have studied it extensively and deeply [2,3,4,5,6]. However, the same ciphertext cannot be classified and stored by SE schemes. A new cryptographic primitive approach emerged as the times required, namely the public key encryption supporting equality test (PKEwET) [7]. In this paper, traceability is introduced into the PKEwET scheme.

### 1.1. Related Work

The PKEwET scheme resolves the problem of data matching in many application environments, such as in cloud computing, health service systems, and IoT. It can compare the consistency of the ciphertexts without the secret key. Recently, the research scope of PKEwET has focused on the three aspects of authorization, security scheme, and efficiency of the PKEwET scheme. Some progress in PKEwET is reviewed as follows:

For the authorization, Tang et al. and Huang et al. proposed PKEwET schemes supporting authorization from the user and ciphertext, respectively [8,9,10,11,12,13]. Then, Ma et al. extended the authorization mechanism to multi-user environments [14]. For more convenient application, Ma et al. proposed four types of authorization policies, namely user level, ciphertext level, user-specific ciphertext level, and ciphertext-to-user level authorization [15]. To simplify the maintenance of public key certificates, Ma et al. introduced the equality test algorithm into an identity-based encryption scheme [16]. For more convenient application to smart cities, Yang et al. proposed a filtered equality test scheme [17]. Later, Wang et al. combined signcryption and an equality test [18]. Recently, Duong et al. presented new lattice-based PKEwET schemes [19].

For the security, in 2016, Lee et al. improved the scheme of Ma, and proposed a new scheme that achieved IND-CCA security [20], and presented an equality test scheme based on the standard model for the first time [21]. In 2017, Wang et al. and Huang et al. proposed a PKEwET scheme from the ciphertext level, and presented the proof of security under the standard model [22,23]. Subsequently, some other PKEwET schemes based on the standard model have been proposed [24,25].

For the efficiency of the PKEwET schemes, Lin et al. and Zhu et al. proposed pairing-free equality test schemes [26,27]. The scheme of Tang was improved upon by Wu et al. [28,29], where the efficiency of computing increased by 36.7% in encryption and by 39.24% in the test algorithm. In 2018, Qu et al. introduced a certificate-less PKEwET scheme [30]. This scheme was improved upon by Elhabob et al. [31,32]. In 2019, Wu et al. combined Zhu et al.’s and Ma et al.’s schemes, and proposed the pairing-free scheme based identity [33]. In the same year, Lee et al. proposed a new PKEwET scheme, from generic assumptions in the random oracle model [34]. To reduce the cost of computing and communication, Ling et al. introduced the group mechanism into a PKEwET algorithm [35].

For convenience in cloud computing of the PKEwET scheme, key-policy attribute-based encryption was introduced by Zhu et al. [36]. In 2018, ciphertext-policy attribute-based encryption was introduced into a PKEwET scheme by Wang et al. [37]. Subsequently, some improvement schemes were put forward [38,39,40,41].

Driven by interests, some users may disclose their own secret keys to non-group users intentionally or unintentionally. However, it is difficult for the malicious user to be tracked down by the system. The problem of key abuse brings great security risks to PKEwET systems. To solve this problem, we introduce a tracking function into a PKEwET system.

### 1.2. Contributions

In this paper, traceability is introduced into a group ID-based encryption (GIBE) scheme. The motivation is to make a GIBE supporting traceability and an equality test function to the ciphertexts. The key contributions can be listed as follows:We show that the GIBE algorithm is unable to compare ciphertexts, and has no equality test function without the secret key sk. To overcome these limitations, we combine the GIBE and PKEwET algorithms. Additionally, all of PKEwET algorithms are untraceable to the encrypted ciphertexts, the idea of traceability is introduced into the PKEwET algorithm, and we propose the traceable GIBE with an equality test scheme (T-GIBEwET).Two types of adversaries are described, and the security of the proposed scheme is proved in details from two types of adversaries. The presented scheme achieves a desirable security. With a trapdoor, the T-GIBEwET scheme can resist OW-CCA security. Without a trapdoor, the T-GIBEwET scheme can resist IND-CCA security.The performance of the T-GIBEwET scheme is discussed. Compared to existing equality test schemes, it is more efficient and more practical in many scenarios.

### 1.3. Outline of This Paper

The rest of the proposal is organized as follows: some preliminaries, some basic definitions, assumptions and the security model are presented in Section 2. The details of the T-GIBEwET scheme are presented in Section 3. The security of the T-GIBEwET scheme is discussed in Section 4. In Section 5, the performance analysis of the T-GIBEwET scheme is represented. Finally, the concluding remarks of this paper are summarized in Section 6.

## 2. Preliminaries

In this section, we present the safety objectives, cryptographic assumptions and security models used in this paper.

### 2.1. Decisional Bilinear Diffie–Hellman Assumption

The proposed scheme is secure under the decisional bilinear Diffie–Hellman assumption.

In this algorithm, the challenger S picks $a,b,c,z\in Z_{p}^{*}$ and flips coin coin∈{0,1} randomly.

If coin=0, S outputs $\left(g,g^{a},g^{b},g^{c},e{\left(g,g\right)}^{z}\right)$.Otherwise, S outputs $\left(g,g^{a},g^{b},g^{c},e{\left(g,g\right)}^{abc}\right)$.

Then, the adversary A gives a guess of coin.

### 2.2. Definition of PKEwET

The PKEwET scheme contains four algorithms [7]:(1)**KeyGen** ($1^{l}$): This procedure randomly selects $x\in Z_{q}^{*}$, and outputs the public/secret key pair $\left(pk=g^{x},sk=x\right)$, where *g* is a generator of *G*.(2)**Encrypt** (M,pk): This procedure selects the numbers $r\in Z_{q}^{*}$ randomly. Then, it outputs the ciphertext CT as follows:Use *r* to compute:
$$C_{1}=g^{r}$$
$$C_{2}=M^{r}$$
$$C_{3}=Hash\left(C_{1},C_{2},g^{xr}\right)⊕\left(M\|r\right)$$Output the ciphertext $CT=(C_{1},C_{2},C_{3})$.(3)**Decrypt**(CT,sk): Given sk and a ciphertext CT, the procedure runs as follows:
$$M\|r=C_{3}⊕Hash\left(C_{1},C_{2},C_{1}^{x}\right)$$If $C_{1}=g^{r}$ and $C_{2}=M^{r}$, output *M*; otherwise, return ⊥.(4)**Test**($CT_{i},CT_{j}$): Given $CT_{i}=\left(C_{i,1},C_{i,2},C_{i,3}\right),CT_{j}=(C_{j,1},C_{j,2},C_{j,3}$ the procedure runs as follows:
$$T_{1}=e\left(C_{i,1},C_{j,2}\right)$$
$$T_{2}=e\left(C_{j,1},C_{i,2}\right)$$Then, check whether $T_{1}=T_{2}$ holds. If yes, it means that $M_{i}=M_{j}$ and output 1. Otherwise, it means that $M_{i}\neq M_{j}$ and output 0.

### 2.3. Group ID-Based Encryption

A group ID-based encryption scheme consists of the following six algorithms [42]:(1)**Setup** (*l*): With the security parameter *l*, this procedure exports system public parameters sp and msk.(2)**KeyGengroup** (sp): With system public parameters sp, this procedure exports the public key and secret key gsk of group users.(3)**Extract** (msk,sp,ID): With a user’s identity $ID\in {\left\{0,1\right\}}^{*}$, this procedure outputs the public key and secret key dk of users.(4)**Join** ($gsk,h_{ID}$): This algorithm is an interactive protocol between the group manager and the prospective user; it takes the group user’s ID as inputs, and outputs the group public key gpk.(5)**Encrypt** ($M,sp,gpk_{i},dk_{ID_{i}},ID_{j}$): This algorithm takes the public keys sp, $gpk_{i}$ of the group manager, $dk_{ID_{i}}$ of the user *i*, and the receiver’s public key $ID_{j}$ and the message *M* as inputs, and outputs a ciphertext CT.(6)**Decrypt** ($CT,gpk,dk_{ID_{j}}$): This algorithm is run by the receiver; it takes the group public key gpk, the receiver’s secret key $dk_{ID_{j}}$, and the ciphertext CT as inputs, and outputs the message *M* or an error symbol ⊥.

### 2.4. System Models

Figure 1 illustrates the system model of T-GIBEwET. The system has four roles: the group manger, the users, the tester, and a trusted third party. The trusted third party generates the private key dk for users. The group manger generates the group public key and group secret key for the group users. The group users encrypt and send the private data to the tester. The tester is authorized and gains a trapdoor gtd.

An integrated T-GIBEwET scheme consists of nine algorithms: **Setup**, **KeyGengroup**, **Extract**, **Join**, **Encrypt**, **Decrypt**, **Trace**, **Auth**, and **Test**.

(1)**Setup** (*l*): With the security parameter *l*, this procedure exports the system public parameters sp and msk.(2)**KeyGengroup** (sp): With system public parameters sp, this procedure exports the public key and secret key gsk of group users.(3)**Extract** (msk,sp,ID): With a user’s identity $ID\in {\left\{0,1\right\}}^{*}$, this procedure outputs the public key and secret key dk of users.(4)**Join** ($gsk,h_{ID}$): This algorithm is an interactive protocol between the group manager and the prospective user; it takes the group user’s ID as inputs, and outputs the group public key gpk.(5)**Encrypt** ($M,sp,gpk_{i},dk_{ID_{i}},ID_{j}$): This algorithm takes the public keys sp and $gpk_{i}$ of the group manager, $dk_{ID_{i}}$ of the user *i*, the receiver’s public key $ID_{j}$, and the message *M* as inputs, and outputs a ciphertext CT.(6)**Decrypt** ($CT,gpk,dk_{ID_{j}}$): This algorithm is run by the receiver, it takes the group public key gpk, the receiver’s secret key $dk_{ID_{j}}$, and the ciphertext CT as inputs, and outputs the message *M* or an error symbol ⊥.(7)**Trace** ($CT,gsk,h_{ID_{i}},gpk$): This algorithm is run by the group manger; it takes group secret key gsk, $h_{ID_{i}}$, gpk, and a ciphertext CT as inputs, and outputs the user’s ID.(8)**Auth** (gsk): This algorithm is run by the group manger, and outputs the group trapdoor gtd.(9)**Test** ($CT_{i},CT_{j},gtd$): This algorithm is run by the tester; it takes the two ciphertexts $CT_{i},CT_{j}$ and gtd as inputs, and outputs 1 or 0.

### 2.5. Security Models

According to different permissions, we show two kinds of adversaries in our proposal.

$Type-\alpha_{1}$ adversary: With a trapdoor, the adversary cannot recover the plaintext after receiving the challenge ciphertext.$Type-\alpha_{2}$ adversary: Without a trapdoor, the adversary cannot tell by which message is $CT^{*}$ encrypted.


**OW-CCA security in T-GIBEwET.**


$Type-\alpha_{1}$ adversary $A_{1}$ and simulator S’s game is played as in Figure 2.

In Figure 2, $O_{1}$ represents the $H_{1}$, $H_{2}$, $H_{3}$, $H_{4}$, and $H_{5}$ queries. $O_{2}\left(ID\right)\overset{△}{=}\mathrm{Extract}\left(msk,ID\right)$, $O_{3}\left(M,ID_{j},gpk,sp,dk_{ID_{i}}\right)\overset{△}{=}\mathrm{Encrypt}\left(M,ID_{j},gpk,sp,dk_{ID_{i}}\right)$, $O_{4}\left(ID,CT\right)\overset{△}{=}\mathrm{Decrypt}$$\left(dk_{ID},CT\right)$, $O_{5}\left(gtd,\cdot \right)\overset{△}{=}\mathrm{Auth}\left(gtd,\cdot \right)$, $O_{6}=O_{1}$, $O_{8}\left(M,ID_{j},gpk,sp,dk_{ID_{i}}\right)=$$O_{3}\left(M,ID_{j},gpk,sp,dk_{ID_{i}}\right)\overset{△}{=}\mathrm{Encrypt}\left(M,ID_{j},gpk,sp,dk_{ID_{i}}\right)$, $O_{10}\left(gtd,\cdot \right)=O_{5}\left(gtd,\cdot \right)\overset{△}{=}\mathrm{Auth}\left(gtd,\cdot \right)$, but
$$O_{7}\left(i\right)=\left\{\begin{array}{ccc} O_{2}\left(i\right) & & i\neq t\\ ⊥ & & otherwise \end{array}\right.$$
and
$$O_{9}\left(i,CT_{i}\right)=\left\{\begin{array}{ccc} O_{4}\left(i,CT_{i}\right) & & CT_{i}\neq CT^{*}\\ ⊥ & & otherwise \end{array}\right.$$

The advantage of $A_{1}$ in the aforementioned game is defined as follows:
$$Adv_{PKEwET-FA,A_{1}}^{OW-CCA}\left(k\right)=\mathrm{Pr}[M_{t}=M_{t}^{*}]$$

As described in Figure 2, $A_{1}$ enjoys $O_{1}$, $O_{2}$, $O_{3}$, $O_{4}$, and $O_{5}$ queries in Phase 1, and S answers all queries truthfully. When $A_{1}$ decides to discontinue queries, S selects a challenge message *M* and generates the challenge ciphertext $CT^{*}$. Then, $A_{1}$ enjoys $O_{6}$, $O_{7}$, $O_{8}$, $O_{9}$, and $O_{10}$ queries as Phase 1, but the condition is that $CT^{*}$ does not appear in $O_{9}$. When $A_{1}$ decides to discontinue queries, $A_{1}$ guesses $M^{{}^{\prime }}$ to S.

**Definition** **1.**
*The T-GIBEwET scheme is OW-CCA security, if all polynomial time and the advantage of $A_{1}$ ($Adv_{T-GIBEwET,A_{1}}^{OW-CCA}\left(l\right)=$
*Pr*$\left[M=M^{{}^{\prime }}\right]$) is negligible in the above game.*



**IND-CCA security in T-GIBEwET.**


$Type-\alpha_{2}$ adversary $A_{2}$ and simulator S’s game is played as in Figure 3.

In Figure 3, $O_{1}$ represents $H_{1}$, $H_{2}$, $H_{3}$, $H_{4}$, and $H_{5}$ queries. $O_{2}\left(ID\right)\overset{△}{=}\mathrm{Extract}\left(msk,ID\right)$, $O_{3}\left(M,ID_{j},gpk,sp,dk_{ID_{i}}\right)\overset{△}{=}\mathrm{Encrypt}\left(M,ID_{j},gpk,sp,dk_{ID_{i}}\right)$, $O_{4}\left(ID,CT\right)\overset{△}{=}\mathrm{Decrypt}$$\left(dk_{ID},CT\right)$, $O_{5}=O_{1}$, $O_{7}\left(M,ID_{j},gpk,sp,dk_{ID_{i}}\right)=O_{3}\left(M,ID_{j},gpk,sp,dk_{ID_{i}}\right)\overset{△}{=}\mathrm{Encrypt}$$\left(M,ID_{j},gpk,sp,dk_{ID_{i}}\right)$, but
$$O_{6}\left(i\right)=\left\{\begin{array}{ccc} O_{2}\left(i\right) & & i\neq t\\ ⊥ & & otherwise \end{array}\right.$$
and
$$O_{8}\left(i,CT_{i}\right)=\left\{\begin{array}{ccc} O_{4}\left(i,CT_{i}\right) & & CT_{i}\neq CT^{*}\\ ⊥ & & otherwise \end{array}\right.$$

The advantage of $A_{2}$ in the aforementioned game is defined as follows:$$Adv_{PKEwET-FA,A_{2}}^{IND-CCA}\left(k\right)=|\mathrm{Pr}[b=b^{*}]-1/2|)$$

As described in Figure 3, $A_{2}$ enjoys $O_{1}$, $O_{2}$, $O_{3}$, and $O_{4}$ queries in Phase 1, and S answers all queries truthfully. When $A_{2}$ decides to discontinue queries, $A_{2}$ selects the two challenge messages $M_{0}$, $M_{1}$. Given $M_{0}$ and $M_{1}$, S outputs $CT^{*}$ based on a random selection of $M_{0}$ and $M_{1}$. Then, $A_{2}$ enjoys $O_{5}$, $O_{6}$, $O_{7}$, and $O_{8}$ queries as Phase 1, but the condition is that $CT^{*}$ does not appear in $O_{8}$. When $A_{2}$ decides to discontinue queries, $A_{2}$ guesses $b^{{}^{\prime }}$ to S.

**Definition** **2.**
*The T-GIBEwET scheme is IND-CCA security, if all polynomial time and the advantage of $A_{2}$ ($Adv_{T-GIBEwET,A_{2}}^{IND-CCA}\left(l\right)=|$
*Pr*$\left[b=b^{{}^{\prime }}]-1/2|\right)$ is negligible in the above game.*


**Definition** **3**(Correctness). *If a T−GIBEwET scheme is correct, for any sp←Setup(l), gsk←KeyGengroup(sp), dk←Extract(msk,sp,ID), $gpk\leftarrow \mathrm{Join}(gsk,h_{ID})$, $CT_{j}\leftarrow \mathrm{Encrypt}$$\left(M,sp,gpk_{i},dk_{ID_{i}},ID_{j}\right)$, $CT_{i}\leftarrow \mathrm{Encrypt}\left(M,sp,gpk_{j},dk_{ID_{j}},ID_{i}\right)$ and gtd←Auth(gsk), the following conditions must be satisfied:*
*(1)* *For any M∈M, Decrypt(Encrypt$\left(M,sp,gpk_{j},dk_{ID_{j}},ID_{i}\right),dk_{ID_{i}})=M$ always holds.**(2)* *For any ciphertexts $CT_{i}$ and $CT_{j}$, if $Decrypt\left(CT_{i},dk_{ID_{i}}\right)=Decrypt\left(CT_{j},dkID_{j}\right)\neq ⊥$, it holds that*$$Test(CT_{i},CT_{j},gtd)=1.$$*(3)* *For any ciphertexts $CT_{i}$ and $CT_{j}$, if $Decrypt\left(CT_{i},dk_{ID_{i}}\right)\neq Decrypt\left(CT_{j},dkID_{j}\right)\neq ⊥$, it holds that*$$Test(CT_{i},CT_{j},gtd)=0.$$


### 2.6. Symbols

In this paragraph, we summarize some symbols used in the proposed scheme. These symbols will assist readers to read and understand the following sections. These symbols are listed in Table 1.

## 3. Our Constructions

This section provides the proposed T-GIBEwET scheme as follows.

(1)**Setup**(*l*): With the security parameter *l*, this procedure exports the system public parameters $sp=(g,G,G_{T},e,g^{s},H_{1},H_{2},H_{3},H_{4},H_{5})$. Choose hash functions: $H_{1}\;:{\left\{0,1\right\}}^{*}\to G^{*}$, $H_{2}\;:G_{T}\to G$, $H_{3}\;:G_{T}\to {\left\{0,1\right\}}^{l+l_{1}}$, $H_{4},H_{5}\;:{\left\{0,1\right\}}^{*}\to {\left\{0,1\right\}}^{l}$; here $l_{1}$ means the length of elements in $Z_{q}$. The master key msk is *s*.(2)**KeyGengroup**(sp): This procedure randomly selects $s_{1},s_{2}\in Z_{q}^{*}$, and outputs the group secret key $gsk=(s_{1},s_{2})$.(3)**Extract**(msk,sp): With a string $ID\in {\left\{0,1\right\}}^{*}$, this procedure outputs the public key and secret key as follows:Outputs a public key $h_{ID}=H_{1}\left(ID\right)\in G^{*}$.Outputs a secret key $dk_{ID}=h_{ID}^{s}$.(4)**Join**($gsk,h_{ID}$): This procedure outputs the group public key $gpk=(h_{ID}^{s_{1}},g^{s_{1}},g^{s_{1}s_{2}})$ for user ID.(5)**Encrypt**($M,sp,gpk_{i},dk_{ID_{i}},ID_{j}$): This procedure selects numbers $r_{1},r_{2}\in Z_{q}^{*}$ randomly. Then, it outputs the ciphertext CT as follows:Use $r_{1},r_{2}$ to compute:
$$C_{1}=h_{ID_{i}}^{s_{1}r_{1}}$$
$$C_{2}=M^{r_{2}}H_{2}\left(U_{1}^{r_{1}}\right)$$
$$C_{3}=g^{r_{1}}$$
$$C_{4}=g^{sr_{2}}$$
$$C_{5}=h_{ID_{i}}^{sr_{1}}$$
$$C_{6}=H_{3}\left(U_{2}^{r_{1}}\right)⊕\left(M\|r_{1}\right)$$
$$C_{7}=H_{5}(C_{1}\|C_{2}\|C_{3}\|C_{4}\|C_{5}\|C_{6}\left\|h_{ID_{i}}^{s}\right)$$
$$C_{8}=H_{4}(C_{1}\|C_{2}\|C_{3}\|C_{4}\|C_{5}\|C_{6}\|C_{7}\left\|M\right\|r_{1}).$$Output the ciphertext $CT=(C_{1},C_{2},C_{3},C_{4},C_{5},C_{6},C_{7},C_{8})$
,where:
$U_{1}=e\left(h_{ID_{i}}^{s},g^{s_{1}s_{2}}\right)$$U_{2}=e\left(h_{ID_{j}},g^{s}\right)$(6)**Decrypt**($CT,dk_{ID_{j}}$): Given $dk_{ID_{j}}$ and a ciphertext CT, the procedure runs as follows:
$$M\|r_{1}=C_{6}⊕H_{3}\left(e\left(C_{3},h_{ID_{j}}^{s}\right)\right)$$If $C_{1}=h_{ID_{i}}^{s_{1}r_{1}}$ and $C_{8}=H_{4}(C_{1}\|C_{2}\|C_{3}\|C_{4}\|C_{5}\|C_{6}\|C_{7}\left\|M\right\|r_{1})$, output *M*; otherwise, return ⊥.(7)**Trace**($CT,dk_{ID_{i}},sp$): Given $dk_{ID_{i}}$, sp and a ciphertext CT, the procedure runs as follows:
$$D_{1}=e\left(g,C_{5}\right)$$
$$D_{2}=e\left(h_{ID_{i}}^{s},C_{3}\right)$$Then, check whether $C_{7}=H_{5}(C_{1}\|C_{2}\|C_{3}\|C_{4}\|C_{5}\|C_{6}\left\|h_{ID_{i}}^{s}\right)$ and $D_{1}=D_{2}$ holds. If yes, it means that CT is encrypted by $ID_{i}$.(8)**The algorithm from the authorization function and test function**:Suppose $CT_{i}$ (resp. $CT_{j}$) is a ciphertext of $ID_{i}$ (resp. $ID_{j}$).**Auth**(gsk): Outputs the group trapdoor $gtd=s_{2}$.**Test**($CT_{i},CT_{j},gtd$):This procedure takes the inputs $CT_{i},CT_{j}$ and gtd and exports as follows:
$$M_{i}^{r_{i,2}}=C_{i,2}/e{\left(C_{i,1},g^{s}\right)}^{s_{2}}$$
$$M_{j}^{r_{j,2}}=C_{j,2}/e{\left(C_{j,1},g^{s}\right)}^{s_{2}}$$Use $M_{i}^{r_{i,2}}$ and $M_{j}^{r_{j,2}}$ to decide whether $e\left(M_{i}^{r_{i,2}},C_{j,4}\right)=e\left(M_{j}^{r_{j,2}},C_{i,4}\right)$. If yes, output 1, which means $M_{i}=M_{j}$. Otherwise, export 0, which means $M_{i}\neq M_{j}$.

**Theorem** **1.**
*According to Definition 3, the above T-GIBEwET scheme is correct.*


**Proof.** We show in turn that the three conditions of Definition 3 are all satisfied.
(1)The first condition is easy to verify.(2)Considering the second condition, for any sp←Setup(l), gsk←KeyGengroup(sp), dk←Extract(msk,sp,ID), $gpk\leftarrow \mathrm{Join}(gsk,h_{ID})$, $CT_{j}\leftarrow \mathrm{Encrypt}\left(M,sp,gpk_{i},dk_{ID_{i}},ID_{j}\right)$, $CT_{i}\leftarrow \mathrm{Encrypt}\left(M,sp,gpk_{j},dk_{ID_{j}},ID_{i}\right)$, the following equalities hold.Given a group trapdoor $gtd=s_{2}$ and two ciphertexts $CT_{i}=\mathrm{Encrypt}$$\left(M_{i},sp,gpk_{j},dk_{ID_{j}},ID_{i}\right)$ and $CT_{j}=\mathrm{Encrypt}\left(M_{j},sp,gpk_{i},dk_{ID_{i}},ID_{j}\right)$, we can compute as follows:
$$C_{i,2}/e{\left(C_{i,1},g^{s}\right)}^{s_{2}}=M_{i}^{r_{i,2}}H_{2}\left(e{\left(h_{ID_{i}}^{s},g^{s_{1}s_{2}}\right)}^{r_{i,1}}\right)/e{\left(h_{ID_{i}}^{s_{1}r_{i,1}},g^{s}\right)}^{s_{2}}$$
$$=M_{i}^{r_{i,2}}H_{2}\left(e{\left(h_{ID_{i}},g\right)}^{ss_{1}s_{2}r_{i,1}}\right)/e{\left(h_{ID_{i}},g\right)}^{ss_{1}s_{2}r_{i,1}}=M_{i}^{r_{i,2}}$$
$$C_{j,2}/e{\left(C_{j,1},g^{s}\right)}^{s_{2}}=M_{j}^{r_{j,2}}H_{2}\left(e{\left(h_{ID_{j}}^{s},g^{s_{1}s_{2}}\right)}^{r_{j,1}}\right)/e{\left(h_{ID_{j}}^{s_{1}r_{j,1}},g^{s}\right)}^{s_{2}}$$
$$=M_{j}^{r_{j,2}}H_{2}\left(e{\left(h_{ID_{j}},g\right)}^{ss_{1}s_{2}r_{j,1}}\right)/e{\left(h_{ID_{j}},g\right)}^{ss_{1}s_{2}r_{j,1}}=M_{j}^{r_{j,2}}$$Use $M_{i}^{r_{i,2}}$ to compute $e\left(M_{i}^{r_{i,2}},C_{j,4}\right)=e\left(M_{i}^{r_{i,2}},g^{sr_{j,2}}\right)=e{\left(M_{i},g\right)}^{sr_{j,2}r_{i,2}}$.Use $M_{j}^{r_{j,2}}$ to compute $e\left(M_{j}^{r_{j,2}},C_{i,4}\right)=e\left(M_{j}^{r_{j,2}},g^{sr_{i,2}}\right)=e{\left(M_{j},g\right)}^{sr_{i,2}r_{j,2}}$. If $M_{i}=M_{j}$, then $e\left(M_{i}^{r_{i,2}},C_{j,4}\right)=e\left(M_{j}^{r_{j,2}},C_{i,4}\right)$, which means $Test(CT_{i},CT_{j},gtd)=1.$(3)As for the third condition, we have the following fact:As in the above calculation, for any message $M_{i}\left(resp.M_{j}\right)$, if $M_{i}\neq M_{j}$, which means $e{\left(M_{i},g\right)}^{sr_{j,2}r_{i,2}}\neq e{\left(M_{j},g\right)}^{sr_{i,2}r_{j,2}}$. Then, $Test(CT_{i},CT_{j},gtd)=0$ holds.
□

## 4. Security Analysis

This section analyzes the security of the scheme and authorization.

**Theorem** **2.**
*For a type-1 adversary, under the random oracle model, the presented T-GIBEwET scheme is OW-CCA secure.*


**Proof.** Let $A_{1}$ be Type-1 adversary breaking the T-GIBEwET scheme in polynomial time. $A_{1}$ makes at most $q_{H_{1}}>0$$H_{1}$-queries, $q_{H_{2}}>0$$H_{2}$-queries, $q_{H_{3}}>0$$H_{3}$-queries, $q_{H_{4}}>0$$H_{4}$-queries, $q_{H_{5}}>0$$H_{5}$-queries, $q_{Key}>0$ key retrieve queries, $q_{Enc}>0$ encryption queries, and $q_{Dec}>0$ decryption queries. We give $CT^{*}$ to the simulator S. The aim of S is to recover the plaintext of $CT^{*}$ with a non-negligible advantage.The game between $A_{1}$ and S is described as follows:
**Game $G_{1.0}$**
**Setup:**S runs the algorithm **Setup**($1^{l}$) to create the system parameters $sp=(g,G,G_{T},g^{s},e,H_{1},H_{2},$ $H_{3},H_{4},H_{5})$, runs the algorithm **KeyGengroup**(sp) to create a group private key $gsk=(s_{1},s_{2})$, runs the algorithm **Join**($gsk,h_{ID}$) to create a group public key $gpk=(h_{ID}^{s_{1}},g^{s_{1}},g^{s_{1}s_{2}})$ for user ID, and runs **Auth**(gsk) to create a group trapdoor $gtd=s_{2}$. Then, S randomly selects $ID_{1},ID_{2}$ as a challenger sender and a challenger receiver, respectively. Then, S gives the public key and $ID_{1},ID_{2}$ to $A_{1}$.Moreover, the challenger S prepares the five hash lists $H_{1},H_{2},H_{3},H_{4},H_{5}$ to record all hash queries and answer the random oracle queries, where all hash lists are empty at the beginning. If the same input is asked multiple times, the same answer will be returned.***Phase 1:***S responds to the queries made by $A_{1}$ in the following ways:
$H_{1}$-query: S maintains a list of 3-tuples $\left(ID_{i},\alpha_{i},x_{i},coin_{i}\right)$ in $H_{1}$. When $A_{1}$, ask for $ID_{i}$ queries, and S runs as follows:
-If the query $ID_{i}$ already in the $H_{1}$ list in the form of $\left(ID_{i},\alpha_{i},x_{i},coin_{i}\right)$, S outputs $H_{1}\left(ID_{i}\right)=\alpha_{i}\in G^{*}$ to $A_{1}$.-Otherwise, S generates $coin_{i}\in \left\{0,1\right\}$ randomly. Then, it outputs as follows:
∗If $coin_{i}=0$, S chooses a random number $x_{i}\in Z_{q}^{*}$ and computes $\alpha_{i}=g^{x_{i}}$ to $A_{1}$.∗Otherwise, S computes $\alpha_{i}=h_{ID_{2}}^{x_{i}}$ to $A_{1}$.-S adds the tuple $\left(ID_{i},\alpha_{i},x_{i},coin_{i}\right)$ into the $H_{1}$ list.$H_{2}$-query: S maintains a list of 2-tuples $\left(\theta_{i},\vartheta_{i}\right)$ in $H_{2}$. S chooses $\vartheta_{i}\in G$ randomly, returns $\vartheta_{i}$ to $A_{1}$, and adds the tuple $\left(\theta_{i},\vartheta_{i}\right)$ to the $H_{2}$ list.$H_{3}$-query: S maintains a list of 2-tuples $\left(\mu_{i},\nu_{i}\right)$ in $H_{3}$. S chooses $\nu_{i}{\left\{0,1\right\}}^{l+l_{1}}$ randomly, returns $\mu_{i}$ to $A_{1}$, and adds the tuple $\left(\mu_{i},\nu_{i}\right)$ to the $H_{3}$ list.$H_{4}$-query: S maintains a list of 2-tuples $\left(\rho_{i},\xi_{i}\right)$ in $H_{4}$. S chooses $\xi_{i}{\left\{0,1\right\}}^{l}$ randomly, returns $\rho_{i}$ to $A_{1}$, and adds the tuple $\left(\rho_{i},\xi_{i}\right)$ to the $H_{4}$ list.$H_{5}$-query: S maintains a list of 2-tuples $\left(\phi_{i},\varphi_{i}\right)$ in $H_{4}$. S chooses $\varphi_{i}{\left\{0,1\right\}}^{l}$ randomly, returns $\phi_{i}$ to $A_{1}$, and adds the tuple $\left(\phi_{i},\varphi_{i}\right)$ to the $H_{5}$ list.Extract Query(ID): When inputting $ID_{i}$, S sends $dk_{ID_{i}}=\alpha_{i}$ to $A_{1}$. If $coin_{i}=1$, it means that $ID\neq ID_{2}$. Then, S sends ⊥ to $A_{1}$.Encryption Query: S runs an encryption algorithm and outputs CT=Encrypt(M,ID,gpk,sp,dk).Decryption queries: With the CT to the decryption query, S returns $M=\mathrm{Decrypt}(CT,dk_{j})$ to $A_{1}$ as follows:
-If $coin_{i}=0$, S uses the private key and outputs the decryption query to $A_{1}$.-Otherwise, S outputs ⊥ to $A_{1}$.Authorization Query: S outputs the group trapdoor $s_{2}$ to $A_{1}$.
***Challenge:***S chooses $M^{*}\subset M$ and $r_{1}^{*},r_{2}^{*}\in {\left\{0,1\right\}}^{l_{1}}$. It then outputs $CT^{*}$ as follows:
$$C_{1}^{*}=h_{ID_{1}}^{s_{1}r_{1}}$$
$$C_{2}^{*}=M^{*r_{2}}H_{2}\left(U_{1}^{r_{1}}\right)$$
$$C_{3}^{*}=g^{r_{1}}$$
$$C_{4}^{*}=g^{sr_{2}}$$
$$C_{5}^{*}=h_{ID_{1}}^{sr_{1}}$$
$$C_{6}^{*}=H_{3}\left(U_{2}^{r_{1}}\right)⊕\left(M^{*}\|r_{1}\right)$$
$$C_{7}^{*}=H_{4}(C_{1}^{*}\|C_{2}^{*}\|C_{3}^{*}\|C_{4}^{*}\|C_{5}^{*}\|C_{6}^{*}\left\|h_{ID_{1}}^{s}\right)$$
$$C_{8}^{*}=H_{4}(C_{1}^{*}\|C_{2}^{*}\|C_{3}^{*}\|C_{4}^{*}\|C_{5}^{*}\|C_{6}^{*}\|C_{7}^{*}\|M^{*}\left\|r_{1}\right).$$The ciphertext $CT^{*}=\left(C_{1}^{*},C_{2}^{*},C_{3}^{*},C_{4}^{*},C_{5}^{*},C_{6}^{*},C_{7}^{*},C_{8}^{*}\right)$ is output, where:

$U_{1}=e\left(h_{ID_{1}}^{s},g^{s_{1}s_{2}}\right)$



$U_{2}=e\left(h_{ID_{2}},g^{s}\right)$

Finally, it sends $CT^{*}$ to $A_{1}$ as the challenge ciphertext.***Phase 2:***$A_{1}$ performs the same queries as in ***Phase 1***; the constraint is that $CT^{*}$ does not appear in the decryption queries.***Guess:***$A_{1}$ outputs $M^{{}^{\prime }}\subset M$.Let $E_{1.0}$ be the event that $M^{{}^{\prime }}=M^{*}$ in **Game $G_{1.0}$**. Then, the advantage is:
$$Adv_{T-GPKE-ET,A_{1}}^{OW-CCA}\left(q_{H_{1}},q_{H_{2}},q_{H_{3}},q_{H_{4}},q_{H_{5}},q_{Extr},q_{Enc},q_{Dec}\right)=Pr\left[E_{1.0}\right]$$
**Game $G_{1.1}$**
**Setup:**S runs the algorithm **Setup**($1^{l}$) to create the system parameters $sp=(g,G,G_{T},g^{s},e,H_{1},H_{2},$ $H_{3},H_{4},H_{5})$, runs the algorithm **KeyGengroup**(sp) to create a group private key $gsk=(s_{1},s_{2})$, runs the algorithm **Join**($gsk,h_{ID}$) to create group public key $gpk=(h_{ID}^{s_{1}},g^{s_{1}},g^{s_{1}s_{2}})$ for user ID, and runs **Auth**(gsk) to create the group trapdoor $gtd=s_{2}$. Then, S randomly selects $ID_{1},ID_{2}$ as a challenger sender and a challenger receiver, respectively. Then, S gives the public key and $ID_{1},ID_{2}$ to $A_{1}$.Moreover, the challenger S prepares the five hash lists $H_{1},H_{2},H_{3},H_{4},H_{5}$ to record all hash queries and answer the random oracle queries, where all hash lists are empty at the beginning. If the same input is asked multiple times, the same answer will be returned.***Phase 1:***S responds to the queries made by $A_{1}$ in the following ways:
$H_{1}$-query (ID), $H_{2}$-query ($\theta_{i}$), $H_{3}$-query ($\mu_{i}$), $H_{4}$-query ($\rho_{i}$), and $H_{5}$-query ($\phi_{i}$) are the same as in **Game $G_{1.0}$**.Extract Query(ID): Same as in **Game $G_{1.0}$**.Encryption Query: S outputs CT to $A_{1}$ as follows: S chooses $r_{1},r_{2}\in {\left\{0,1\right\}}^{l_{1}}$ randomly, and performs the $H_{1}$-query($ID_{i}$), $H_{1}$-query($ID_{j}$) to obtain $\alpha_{i}$, $\alpha_{j}$, the $H_{2}$-query($e{\left(C_{1},g^{s}\right)}^{s_{2}}$) to obtain $\vartheta_{i}$, the $H_{3}$-query($e{\left(\alpha_{j},g^{s}\right)}^{r_{1}}$) to obtain $\nu_{i}$, the $H_{5}$-query($C_{1}\|C_{2}\|C_{3}\|C_{4}\|C_{5}\left\|C_{6}\right\|h_{ID_{i}}^{s}$) to obtain $\varphi_{i}$. and the $H_{4}$-query($C_{1}\|C_{2}\|C_{3}\|C_{4}\|C_{5}\|C_{6}\|C_{7}\left\|M\right\|r_{1}$) to obtain $\xi_{i}$.
$$C_{1}=\alpha_{i}^{s_{1}r_{1}}$$
$$C_{2}=M^{r_{2}}\vartheta_{i}$$
$$C_{3}=g^{r_{2}}$$
$$C_{4}=g^{sr_{2}}$$
$$C_{5}=\alpha_{j}^{sr_{1}}$$
$$C_{6}=\nu_{i}⊕\left(M\|r_{1}\right)$$
$$C_{7}=\varphi_{i}.$$
$$C_{8}=\xi_{i}.$$S adds $\left(e{\left(C_{1},g^{s}\right)}^{s_{2}},\vartheta_{i}\right)$ to the $H_{2}$ list, adds $\left(e{\left(\alpha_{j},g^{s}\right)}^{r_{1}},\nu_{i}\right)$ to the $H_{3}$ list, adds ($C_{1}\|C_{2}\|C_{3}\|C_{4}\|C_{5}\|C_{6}\|C_{7}\left\|M\right\|r_{1},\xi_{i}$) to the $H_{4}$ list, and adds ($C_{1}\|C_{2}\|C_{3}\|C_{4}\|C_{5}\left\|C_{6}\right\|h_{ID_{i}}^{s},\varphi_{i}$) to the $H_{5}$ list.Decryption queries: With the CT to the decryption query, S returns $M=\mathrm{Decrypt}(CT,dk_{j})$ to $A_{1}$ as follows: S performs the $H_{3}\left(e{\left(\alpha_{j},g^{s}\right)}^{r_{1}}\right)$ to obtain answer $\nu_{i}$, and performs the $H_{4}$-query($C_{1}\|C_{2}\|C_{3}\|C_{4}\|C_{5}\|C_{6}\|C_{7}\left\|M\right\|r_{1}$) to obtain answer $\xi_{i}$. Then, S performs
$$M\|r_{1}=C_{6}⊕\nu_{i}.$$Then, it verifies $C_{1}=\alpha_{i}^{s_{1}r_{1}}$ and $C_{8}=\xi_{i}$. If the verification fails, it returns ⊥. Otherwise, S outputs *M* to $A_{1}$.Authorization Query: Same as in **Game $G_{1.0}$**.
***Challenge:***S chooses $M^{*}\subset M$, $W\in {\left\{0,1\right\}}^{l+l_{1}}$ and $r_{1},r_{2}\in {\left\{0,1\right\}}^{l_{1}}$. Then, it outputs $CT^{*}$ as follows:
$$C_{1}^{*}=h_{ID_{1}}^{s_{1}r_{1}}$$
$$C_{2}^{*}=M^{*r_{2}}H_{2}\left(U_{1}^{r_{1}}\right)$$
$$C_{3}^{*}=g^{r_{1}}$$
$$C_{4}^{*}=g^{sr_{2}}$$
$$C_{5}^{*}=h_{ID_{1}}^{sr_{1}}$$
$$C_{6}^{*}=W⊕\left(M^{*}\|r_{1}\right)$$
$$C_{7}^{*}=H_{5}(C_{1}^{*}\|C_{2}^{*}\|C_{3}^{*}\|C_{4}^{*}\|C_{5}^{*}\|C_{6}^{*}\left\|h_{ID_{i}}^{s}\right)$$
$$C_{8}^{*}=H_{4}(C_{1}^{*}\|C_{2}^{*}\|C_{3}^{*}\|C_{4}^{*}\|C_{5}^{*}\|C_{6}^{*}\|C_{7}^{*}\|M^{*}\left\|r_{1}\right)$$
where $U_{1}=e\left(h_{ID_{1}}^{s},g^{s_{1}s_{2}}\right)$. It outputs the ciphertext $CT^{*}=\left(C_{1}^{*},C_{2}^{*},C_{3}^{*},C_{4}^{*},C_{5}^{*},C_{6}^{*},C_{7}^{*},C_{8}^{*}\right)$, and adds $\left(e{\left(h_{ID_{2}},g^{s}\right)}^{r_{1}},W\right)$ into $H_{3}$.Finally, it sends $CT^{*}$ to $A_{1}$ as the challenge ciphertext.***Phase 2:***$A_{1}$ performs the same queries as in ***Phase 1***, where the constraint is that $CT^{*}$ does not appear in the decryption queries.***Guess:***$A_{1}$ outputs $M^{{}^{\prime }}\subset M$.Let $E_{1.1}$ be the event that $M^{{}^{\prime }}=M^{*}$ in **Game $G_{1.1}$**. Then, the advantage is:
$$Pr\left[E_{1.1}\right]=Pr\left[E_{1.0}\right].$$
**Game $G_{1.2}$**
**Setup:**S runs the algorithm **Setup**($1^{l}$) to create the system parameters $sp=(g,G,G_{T},g^{s},e,H_{1},H_{2},$ $H_{3},H_{4},H_{5})$, runs the algorithm **KeyGengroup**(sp) to create a group private key $gsk=(s_{1},s_{2})$, runs the algorithm **Join**($gsk,h_{ID}$) to create the group public key $gpk=(h_{ID}^{s_{1}},g^{s_{1}},g^{s_{1}s_{2}})$ for user ID, and runs **Auth**(gsk) to create the group trapdoor $gtd=s_{2}$. Then, S randomly select $ID_{1},ID_{2}$ as a challenger sender and a challenger receiver, respectively. Then, S gives the public key and $ID_{1},ID_{2}$ to $A_{1}$.Moreover, the challenger S prepares the five hash lists $H_{1},H_{2},H_{3},H_{4},,H_{5}$ to record all hash queries and answer the random oracle queries, where all hash lists are empty at the beginning. If the same input is asked multiple times, the same answer will be returned.***Phase 1:***S responds to the queries made by $A_{1}$ in the following ways:
The $H_{1}$-query(ID), $H_{2}$-query($\theta_{i}$), $H_{5}$-query($\phi_{i}$), and $H_{4}$-query($\rho_{i}$) are the same as in **Game $G_{1.1}$**.The $H_{3}$-query($\mu_{i}$) is the same as in **Game $G_{1.1}$**, except that $A_{1}$ asks $e(C_{3},h_{ID_{2}}^{s})$.Extract Query(ID): Same as in **Game $G_{1.1}$**.Encryption Query: Same as in **Game $G_{1.1}$**.Decryption Queries: Same as in **Game $G_{1.1}$**.Authorization Query: Same as in **Game $G_{1.1}$**.
***Challenge:***S chooses $M^{*}\subset M$, $W^{*}\in {\left\{0,1\right\}}^{l+l_{1}}$ and $r_{1},r_{2}\in {\left\{0,1\right\}}^{l_{1}}$. Then, it outputs $CT^{*}$ as follows:
$$C_{1}^{*}=h_{ID_{1}}^{s_{1}r_{1}}$$
$$C_{2}^{*}=M^{*r_{2}}H_{2}\left(U_{1}^{r_{1}}\right)$$
$$C_{3}^{*}=g^{r_{1}}$$
$$C_{4}^{*}=g^{sr_{2}}$$
$$C_{5}^{*}=h_{ID_{1}}^{sr_{1}}$$
$$C_{6}^{*}=W^{*}$$
$$C_{7}^{*}=H_{5}(C_{1}^{*}\|C_{2}^{*}\|C_{3}^{*}\|C_{4}^{*}\|C_{5}^{*}\|C_{6}^{*}\left\|h_{ID_{i}}^{s}\right).$$
$$C_{8}^{*}=H_{4}(C_{1}^{*}\|C_{2}^{*}\|C_{3}^{*}\|C_{4}^{*}\|C_{5}^{*}\|C_{6}^{*}\|C_{7}^{*}\|M^{*}\left\|r_{1}\right).$$
where $U_{1}=e\left(h_{ID_{1}}^{s},g^{s_{1}s_{2}}\right)$. It outputs the ciphertext $CT^{*}=\left(C_{1}^{*},C_{2}^{*},C_{3}^{*},C_{4}^{*},C_{5}^{*},C_{6}^{*},C_{7}^{*},C_{8}^{*}\right)$, and adds $(e{\left(h_{ID_{2}},g^{s}\right)}^{r_{1}},W^{*}⊕\left(M^{*}\|r_{1}\right))$ into $H_{3}$.Finally, it sends $CT^{*}$ to $A_{1}$ as the challenge ciphertext.***Phase 2:***$A_{1}$ performs the same queries as in ***Phase 1***, whereqthe constraint is that $CT^{*}$ does not appear in the decryption Queries, and if $A_{1}$ asks for the decryption of $CT^{*}=\left(C_{1}^{*},C_{2}^{*},C_{3}^{*},C_{4}^{*},C_{5}^{*},C_{6}^{{}^{\prime }},C_{7}^{*},C_{8}^{*}\right)$, where $C_{6}^{{}^{\prime }}\neq C_{6}^{*}$, S outputs ⊥.***Guess:***$A_{1}$ outputs $M^{{}^{\prime }}\subset M$. □

Let $E_{1.2}$ be the event that $M^{{}^{\prime }}=M^{*}$ in **Game $G_{1.2}$**.

Because $C_{6}^{{}^{\prime }}$ is a random value in **Game $G_{1.1}$** and **Game $G_{1.2}$**, the challenge ciphertexts generated in **Game $G_{1.1}$** and **Game $G_{1.2}$** follow the same distribution. Therefore, if the event $E_{1}$ does not occur, **Game $G_{1.2}$** is identical to **Game $G_{1.1}$**, and we can figure out
$$|Pr\left[E_{1.2}\right]-Pr\left[E_{1.1}\right]|\leq Pr\left[E_{1}\right].$$

Next, we show that the probability of event $E_{1}$ occurring in **Game $G_{1.2}$** is negligible.

**Lemma** **1.***When the C-BDH problem is intractable, there is a negligible probability that the event $E_{1}$ happens in ***Game**$G_{1.2}$.

**Proof.** Suppose that $Pr[E_{1}]$ is non-negligible; we can construct a simulator S to break the C-BDH assumption by using $A_{1}$’s attacks. With the tuple $\left(e,G,G_{T},g,g^{a},g^{c},g^{d}\right)$, the aim is to obtain $e{\left(g,g\right)}^{acd}$.**Setup:**S randomly selects $ID_{1},ID_{2}$ as a challenger sender and a challenger receiver, respectively. Then, S gives the public key and $ID_{1},ID_{2}$ to $A_{1}$. S runs the algorithm **Setup**($1^{l}$) to create the system parameters $sp=(g,G,G_{T},g^{s},e,H_{1},H_{2},$ $H_{3},H_{4},H_{5})$, runs the algorithm **KeyGengroup**(sp) to create a group private key $gsk=(s_{1},s_{2})$, runs the algorithm **Join**($gsk,h_{ID}$) to create the group public key $gpk=(h_{ID}^{s_{1}},g^{s_{1}},g^{s_{1}s_{2}})$ for user ID, and runs **Auth**(gsk) to create the group trapdoor $gtd=s_{2}$.**Phase 1:**S responds to the queries made by $A_{1}$ in the following ways:
$H_{1}$-query(ID), $H_{2}$-query($\theta_{i}$), $H_{5}$-query($\phi_{i}$), and $H_{4}$-query($\rho_{i}$) are same as in **Game $G_{1.1}$**.$H_{3}$-query($\mu_{i}$) is same as in **Game $G_{1.1}$**, except that $A_{1}$ asks $e(C_{3},h_{ID_{2}}^{s})$Extract Query(ID): Same as in **Game $G_{1.1}$**.Encryption Query: Same as in **Game $G_{1.1}$**, except that for the query $\left(ID_{2},*,*\right)$, S selects $r_{1},r_{2}\in {\left\{0,1\right\}}^{l_{1}}$ randomly and outputs a ciphertext $CT=(C_{1},C_{2},C_{3},C_{4},C_{5},C_{6},C_{7},C_{8})$ as follows:S performs the $H_{1}$-query($ID_{i}$) and $H_{1}$-query($ID_{j}$) to obtain $\alpha_{i}$ and $\alpha_{j}$, respectively, the $H_{2}$-query($e{\left(C_{1},g^{s}\right)}^{s_{2}}$) to obtain $\vartheta_{i}$, the $H_{3}$-query($e{\left(\alpha_{j},g^{s}\right)}^{r_{1}}$) to obtain $\nu_{i}$, the $H_{5}$-query($C_{1}\|C_{2}\|C_{3}\|C_{4}\|C_{5}\left\|C_{6}\right\|h_{ID_{i}^{s}}$) to obtain $\varphi_{i}$, and the $H_{4}$-query($C_{1}\|C_{2}\|C_{3}\|C_{4}\|C_{5}\|C_{6}\|C_{7}\left\|M\right\|r_{1}$) to obtain $\xi_{i}$.
$$C_{1}=\alpha_{i}^{s_{1}r_{1}}$$
$$C_{2}=M^{r_{2}}\vartheta_{i}$$
$$C_{3}=g^{r_{2}}$$
$$C_{4}=g^{sr_{2}}$$
$$C_{5}=\alpha_{j}^{sr_{2}}$$
$$C_{6}=\nu_{i}⊕\left(M\|r_{1}\right)$$
$$C_{7}=\varphi_{i}.$$
$$C_{8}=\xi_{i}.$$S adds $\left(e{\left(C_{1},g^{s}\right)}^{s_{2}},\vartheta_{i}\right)$ to the $H_{2}$ list, adds $\left(e{\left(\alpha_{j},g^{s}\right)}^{r_{1}},\nu_{i}\right)$ to the $H_{3}$ list, and adds ($C_{1}\|C_{2}\|C_{3}\|C_{4}\|C_{5}\|C_{6}\|C_{7}\left\|M\right\|r_{1},\xi_{i}$) to the $H_{4}$ list.Decryption queries: Same as in **Game $G_{1.1}$**.Authorization Query: Same as in **Game $G_{1.1}$**.
***Challenge:***S chooses $M^{*}\subset M$, $W^{*}\in {\left\{0,1\right\}}^{l+l_{1}}$ and $r_{1},r_{2}\in {\left\{0,1\right\}}^{l_{1}}$. Then, it outputs $CT^{*}$ as follows:
$$C_{1}^{*}=h_{ID_{1}}^{s_{1}r_{1}}$$
$$C_{2}^{*}=M^{*r_{2}}H_{2}\left(U_{1}^{r_{1}}\right)$$
$$C_{3}^{*}=g^{r_{1}}$$
$$C_{4}^{*}=g^{sr_{2}}$$
$$C_{5}^{*}=h_{ID_{1}}^{sr_{1}}$$
$$C_{6}^{*}=W^{*}$$
$$C_{7}^{*}=H_{5}(C_{1}^{*}\|C_{2}^{*}\|C_{3}^{*}\|C_{4}^{*}\|C_{5}^{*}\|C_{6}^{*}\left\|h_{ID_{1}}^{s}\right).$$
$$C_{8}^{*}=H_{4}(C_{1}^{*}\|C_{2}^{*}\|C_{3}^{*}\|C_{4}^{*}\|C_{5}^{*}\|C_{6}^{*}\|C_{7}^{*}\|M^{*}\left\|r_{1}\right).$$
where $U_{1}=e\left(h_{ID_{1}}^{s},g^{s_{1}s_{2}}\right)$. It outputs the ciphertext $CT^{*}=\left(C_{1}^{*},C_{2}^{*},C_{3}^{*},C_{4}^{*},C_{5}^{*},C_{6}^{*},C_{7}^{*},C_{8}^{*}\right)$, and adds $(e{\left(h_{ID_{2}},g^{s}\right)}^{r_{1}},W^{*}⊕\left(M^{*}\|r_{1}\right))$ into $H_{3}$.Finally, it sends $CT^{*}$ to $A_{1}$ as the challenge ciphertext.***Phase 2:***$A_{1}$ performs the same queries as in ***Phase 1***; the constraint is that $CT^{*}$ does not appear in the decryption queries, and if $A_{1}$ asks for the decryption of $CT^{*}=\left(C_{1}^{*},C_{2}^{*},C_{3}^{*},C_{4}^{*},C_{5}^{*},C_{6}^{{}^{\prime }},C_{7}^{*},C_{8}^{*}\right)$, where $C_{6}^{{}^{\prime }}\neq C_{6}^{*}$, S outputs ⊥.***Guess:***$A_{1}$ outputs $M^{{}^{\prime }}\subset M$. □

**Theorem** **3.**
*Under the random oracle model, the proposed T-GIBEwET scheme is IND-CCA secure against a type-2 adversary.*


**Proof.** Let $A_{2}$ be a type-2 adversary breaking the T-GIBEwET scheme in polynomial time. $A_{2}$ makes at most $q_{H_{1}}>0$$H_{1}$-queries, $q_{H_{2}}>0$$H_{2}$-queries, $q_{H_{3}}>0$$H_{3}$-queries, $q_{H_{4}}>0$$H_{4}$-queries, $q_{H_{5}}>0$$H_{5}$-queries, $q_{Key}>0$ key retrieve queries, $q_{Enc}>0$ encryption queries, and $q_{Dec}>0$ decryption queries. We give $CT^{*}$ to the simulator S. The aim of S is to recover the plaintext of $CT^{*}$ with a non-negligible advantage.The game between $A_{2}$ and S is described as follows:
**Game $G_{2.0}$**
**Setup:**S runs the algorithm **Setup**($1^{l}$) to create the system parameters $sp=(g,G,G_{T},g^{s},e,H_{1},H_{2},$ $H_{3},H_{4},H_{5})$, runs the algorithm **KeyGengroup**(sp) to create a group private key $gsk=(s_{1},s_{2})$, runs the algorithm **Join**($gsk,h_{ID}$) to create the group public key $gpk=(h_{ID}^{s_{1}},g^{s_{1}},g^{s_{1}s_{2}})$ for user ID, and runs **Auth**(gsk) to create the group trapdoor $gtd=s_{2}$. Then, S randomly selects $ID_{1},ID_{2}$ as a challenger sender and a challenger receiver, respectively. Then, S gives the public key and $ID_{1},ID_{2}$ to $A_{2}$.Moreover, the challenger S prepares the five hash lists $H_{1},H_{2},H_{3},H_{4},H_{5}$ to record all hash queries and answer the random oracle queries, where all hash lists are empty at the beginning. If the same input is asked multiple times, the same answer will be returned.***Phase 1:***S responds to the queries made by $A_{2}$ in the following ways:
$H_{1}$-query: S maintains a list of 3-tuples $\left(ID_{i},\alpha_{i},x_{i},coin_{i}\right)$ in $H_{1}$. When $A_{2}$ asks for $ID_{i}$ queries, S runs as follows:
-If the query $ID_{i}$ is already in the $H_{1}$ list in the form of $\left(ID_{i},\alpha_{i},x_{i},coin_{i}\right)$, S outputs $H_{1}\left(ID_{i}\right)=\alpha_{i}\in G^{*}$ to $A_{2}$.-Otherwise, S generates $coin_{i}\in \left\{0,1\right\}$ randomly. Then, it outputs as follows:
∗If $coin_{i}=0$, S chooses a random number $x_{i}\in Z_{q}^{*}$ and computes $\alpha_{i}=g^{x_{i}}$ to $A_{2}$.∗Otherwise, S computes $\alpha_{i}=h_{ID_{2}}^{x_{i}}$ to $A_{2}$.S adds the tuple $\left(ID_{i},\alpha_{i},x_{i},coin_{i}\right)$ into the $H_{1}$ list.$H_{2}$-query: S maintains a list of 2-tuples $\left(\theta_{i},\vartheta_{i}\right)$ in $H_{2}$. S chooses $\vartheta_{i}\in G$ randomly, puts out $\vartheta_{i}$ to $A_{2}$ and adds the tuple $\left(\theta_{i},\vartheta_{i}\right)$ to the $H_{2}$ list.$H_{3}$-query: S maintains a list of 2-tuples $\left(\mu_{i},\nu_{i}\right)$ in $H_{3}$. S chooses $\nu_{i}{\left\{0,1\right\}}^{l+l_{1}}$ randomly, puts out $\mu_{i}$ to $A_{2}$ and adds the tuple $\left(\mu_{i},\nu_{i}\right)$ to the $H_{3}$ list.$H_{4}$-query: S maintains a list of 2-tuples $\left(\rho_{i},\xi_{i}\right)$ in $H_{4}$. S chooses $\xi_{i}{\left\{0,1\right\}}^{l}$ randomly, puts out $\rho_{i}$ to $A_{2}$ and adds the tuple $\left(\rho_{i},\xi_{i}\right)$ to the $H_{4}$ list.$H_{5}$-query: S maintains a list of 2-tuples $\left(\phi_{i},\varphi_{i}\right)$ in $H_{4}$. S chooses $\varphi_{i}{\left\{0,1\right\}}^{l}$ randomly, returns $\phi_{i}$ to $A_{1}$ and adds the tuple $\left(\phi_{i},\varphi_{i}\right)$ to the $H_{5}$ list.Extract Query(ID): On input of the $ID_{i}$, S sends $dk_{ID_{i}}=\alpha_{i}$ to $A_{2}$. If $coin_{i}=1$, which means $ID\neq ID_{2}$, then S sends ⊥ to $A_{2}$.Encryption Query: S runs the encryption algorithm and outputs CT=Encrypt(M,gpk,dk,sp).Decryption queries: With the CT in the decryption query, S returns $M=\mathrm{Decrypt}(CT,dk_{j})$ to $A_{2}$ as follows:
-If $coin_{i}=0$, S uses the private key and outputs the decryption query to $A_{2}$.-Otherwise, S outputs ⊥ to $A_{2}$.Authorization Query: It is not allowed.
***Challenge:***$A_{2}$ chooses $M_{0},M_{1}\subset M$ randomly and sends them to S. Then, S takes b∈{0,1} and $r_{1}^{*},r_{2}^{*}\in {\left\{0,1\right\}}^{l_{1}}$. It then outputs $CT^{*}$ as follows:
$$C_{1}^{*}=h_{ID_{1}}^{s_{1}r_{1}}$$
$$C_{2}^{*}=M_{b}^{r_{2}}H_{2}\left(U_{1}^{r_{1}}\right)$$
$$C_{3}^{*}=g^{r_{1}}$$
$$C_{4}^{*}=g^{sr_{2}}$$
$$C_{5}^{*}=h_{ID_{1}}^{sr_{1}}$$
$$C_{6}^{*}=H_{3}\left(U_{2}^{r_{1}}\right)⊕\left(M_{b}\|r_{1}\right)$$
$$C_{7}^{*}=H_{5}(C_{1}^{*}\|C_{2}^{*}\|C_{3}^{*}\|C_{4}^{*}\|C_{5}^{*}\|C_{6}^{*}\left\|h_{ID_{1}}^{s}\right).$$
$$C_{8}^{*}=H_{4}(C_{1}^{*}\|C_{2}^{*}\|C_{3}^{*}\|C_{4}^{*}\|C_{5}^{*}\|C_{6}^{*}\|C_{7}^{*}\|M_{b}\left\|r_{1}\right).$$Output the ciphertext $CT^{*}=\left(C_{1}^{*},C_{2}^{*},C_{3}^{*},C_{4}^{*},C_{5}^{*},C_{6}^{*},C_{7}^{*},C_{8}^{*}\right)$,
where:
$U_{1}=e\left(h_{ID_{1}}^{s},g^{s_{1}s_{2}}\right)$$U_{2}=e\left(h_{ID_{2}},g^{s}\right)$Finally, it sends $CT^{*}$ to $A_{2}$ as the challenge ciphertext.***Phase 2:***$A_{2}$ performs the same queries as in ***Phase 1***, where the constraint are as follows:
$CT^{*}$ does not appear in the decryption queries.In the authorization query, all of the group users cannot be authorized.
***Guess:***$A_{2}$ outputs $b^{*}\in \left\{0,1\right\}$.Let $E_{2.0}$ be the event that $b=b^{*}$ in **Game $G_{2.0}$**. Then, the advantage is:
$$Adv_{T-GPKE-ET,A_{2}}^{OW-CCA}\left(q_{H_{1}},q_{H_{2}},q_{H_{3}},q_{H_{4}},q_{H_{5}},q_{Extr},q_{Enc},q_{Dec}\right)=Pr\left[E_{2.0}\right]$$
**Game $G_{2.1}$**
**Setup:**S runs the algorithm **Setup**($1^{l}$) to create the system parameters $sp=(g,G,G_{T},g^{s},e,H_{1},H_{2},$ $H_{3},H_{4})$, runs the algorithm **KeyGengroup**(sp) to create a group private key $gsk=(s_{1},s_{2})$, runs the algorithm **Join**($gsk,h_{ID}$) to create the group public key $gpk=(h_{ID}^{s_{1}},g^{s_{1}},g^{s_{1}s_{2}})$ for user ID, and runs **Auth**(gsk) to create the group trapdoor $gtd=s_{2}$. Then, S randomly selects $ID_{1},ID_{2}$ as a challenger sender and a challenger receiver, respectively. Then, S gives the public key and $ID_{1},ID_{2}$ to $A_{2}$.Moreover, the challenger S prepares the four hash lists $H_{1},H_{2},H_{3},H_{4}$ to record all hash queries and answer the random oracle queries, where all hash list are empty at the beginning. If the same input is asked multiple times, the same answer will be returned.***Phase 1:***S responds to the queries made by $A_{2}$ in the following ways:
$H_{1}$-query(ID), $H_{2}$-query($\theta_{i}$), $H_{3}$-query($\mu_{i}$), $H_{5}$-query($\phi_{i}$), and $H_{4}$-query($\rho_{i}$) are the same as in **Game $G_{2.0}$**.Extract Query(ID): Same as in **Game $G_{2.0}$**.Encryption Query: S outputs CT to $A_{2}$ as follows:S chooses $r_{1},r_{2}\in {\left\{0,1\right\}}^{l_{1}}$ randomly, and performs the $H_{1}$-query($ID_{i}$) and $H_{1}$-query($ID_{j}$) to obtain $\alpha_{i}$ and $\alpha_{j}$, respectively, the $H_{2}$-query($e{\left(C_{1},g^{s}\right)}^{s_{2}}$) to obtain $\vartheta_{i}$, the $H_{3}$- query($e{\left(\alpha_{j},g^{s}\right)}^{r_{1}}$) to obtain $\nu_{i}$, the $H_{5}$-query($C_{1}\|C_{2}\|C_{3}\|C_{4}\|C_{5}\left\|C_{6}\right\|h_{ID_{i}}^{s}$) to obtain $\varphi_{i}$, and the $H_{4}$-query($C_{1}\|C_{2}\|C_{3}\|C_{4}\|C_{5}\|C_{6}\|C_{7}\left\|M\right\|r_{1}$) to obtain $\xi_{i}$.
$$C_{1}=\alpha_{i}^{s_{1}r_{1}}$$
$$C_{2}=M^{r_{2}}\vartheta_{i}$$
$$C_{3}=g^{r_{2}}$$
$$C_{4}=g^{sr_{2}}$$
$$C_{5}=\alpha_{j}^{sr_{1}}$$
$$C_{6}=\nu_{i}⊕\left(M\|r_{1}\right)$$
$$C_{7}=\varphi_{i}$$
$$C_{8}=\xi_{i}.$$S adds $\left(e{\left(C_{1},g^{s}\right)}^{s_{2}},\vartheta_{i}\right)$ to the $H_{2}$ list, adds $\left(e{\left(\alpha_{j},g^{s}\right)}^{r_{1}},\nu_{i}\right)$ to the $H_{3}$ list, adds ($C_{1}\|C_{2}\|C_{3}\|C_{4}\|C_{5}\|C_{6}\|C_{7}\left\|M\right\|r_{1},\xi_{i}$) to the $H_{4}$ list, and adds ($C_{1}\|C_{2}\|C_{3}\|C_{4}\|C_{5}\left\|C_{6}\right\|h_{ID_{i}}^{s},\varphi_{i}$) to the $H_{5}$ list.Decryption queries: With the CT to the decryption query, S returns $M=\mathrm{Decrypt}(CT,sk_{j})$ to $A_{2}$ as follows: S performs the $H_{3}\left(e{\left(\alpha_{j},g^{s}\right)}^{r_{1}}\right)$ to obtain answer $\nu_{i}$, and performs the $H_{4}$-query($C_{1}\|C_{2}\|C_{3}\|C_{4}\|C_{5}\|C_{6}\|C_{7}\left\|M\right\|r_{1}$) to obtain answer $\xi_{i}$. Then, S performs
$$M\|r_{1}=C_{6}⊕\nu_{i}.$$Then, $C_{1}=\alpha_{i}^{s_{1}r_{1}}$ and $C_{8}=\xi_{i}$ are verified. If the verification fails, it returns ⊥. Otherwise, S outputs *M* to $A_{2}$.Authorization Query: It is not allowed.
***Challenge:***$A_{2}$ chooses $M_{0},M_{1}\subset M$ randomly and sends them to S. Then, S takes b∈{0,1}, $W\in {\left\{0,1\right\}}^{l+l_{1}}$ and $r_{1}^{*},r_{2}^{*}\in {\left\{0,1\right\}}^{l_{1}}$. It then outputs $CT^{*}$ as follows:
$$C_{1}^{*}=h_{ID_{1}}^{s_{1}r_{1}}$$
$$C_{2}^{*}=M_{b}^{r_{2}}H_{2}\left(U_{1}^{r_{1}}\right)$$
$$C_{3}^{*}=g^{r_{1}}$$
$$C_{4}^{*}=g^{sr_{2}}$$
$$C_{5}^{*}=h_{ID_{1}}^{sr_{1}}$$
$$C_{6}^{*}=W⊕\left(M_{b}\|r_{1}\right)$$
$$C_{7}^{*}=H_{5}(C_{1}^{*}\|C_{2}^{*}\|C_{3}^{*}\|C_{4}^{*}\|C_{5}^{*}\|C_{6}^{*}\left\|h_{ID_{1}}^{s}\right)$$
$$C_{8}^{*}=H_{4}(C_{1}^{*}\|C_{2}^{*}\|C_{3}^{*}\|C_{4}^{*}\|C_{5}^{*}\|C_{6}^{*}\|C_{7}^{*}\|M_{b}\left\|r_{1}\right)$$
where $U_{1}=e\left(h_{ID_{1}}^{s},g^{s_{1}s_{2}}\right)$. It outputs the ciphertext $CT^{*}=\left(C_{1}^{*},C_{2}^{*},C_{3}^{*},C_{4}^{*},C_{5}^{*},C_{6}^{*},C_{7}^{*},C_{8}^{*}\right)$, and adds $\left(e{\left(h_{ID_{2}},g^{s}\right)}^{r_{1}},W\right)$ into $H_{3}$.Finally, it sends $CT^{*}$ to $A_{2}$ as the challenge ciphertext.***Phase 2:***$A_{2}$ performs the same queries as in ***Phase 1***; the constraint are as follows:
$CT^{*}$ does not appear in the decryption queries.In the authorization query, all of the group users cannot be authorized.
***Guess:***$A_{2}$ outputs $b^{*}\in \left\{0,1\right\}$.Let $E_{2.1}$ be the event that $b=b^{*}$ in **Game $G_{2.1}$**. Then, the advantage is
$$Pr\left[E_{2.1}\right]=Pr\left[E_{2.0}\right].$$
**Game $G_{2.2}$**
**Setup:**S runs the algorithm **Setup**($1^{l}$) to create the system parameters $sp=(g,G,G_{T},g^{s},e,H_{1},H_{2},$ $H_{3},H_{4},H_{5})$, runs the algorithm **KeyGengroup**(sp) to create a group private key $gsk=(s_{1},s_{2})$, runs the algorithm **Join**($gsk,h_{ID}$) to create the group public key $gpk=(h_{ID}^{s_{1}},g^{s_{1}},g^{s_{1}s_{2}})$ for user ID, and runs **Auth**(gsk) to create the group trapdoor $gtd=s_{2}$. Then, S randomly selects $ID_{1},ID_{2}$ as a challenger sender and a challenger receiver, respectively. Then, S gives the public key and $ID_{1},ID_{2}$ to $A_{2}$.Moreover, the challenger S prepares the five hash lists $H_{1},H_{2},H_{3},H_{4},H_{5}$ to record all hash queries and answers the random oracle queries, where all hash list are empty at the beginning. If the same input is asked multiple times, the same answer will be returned.***Phase 1:***S responds to the queries made by $A_{2}$ in the following ways:
$H_{1}$-query(ID), $H_{2}$-query($\theta_{i}$), $H_{5}$-query($\phi_{i}$), and $H_{4}$-query($\rho_{i}$) are the same as in **Game $G_{2.1}$**.$H_{3}$-query($\mu_{i}$) is the same as in **Game $G_{2.1}$**, except that $A_{2}$ asks $e(C_{3},h_{ID_{2}}^{s})$.Extract Query(ID): Same as in **Game $G_{2.1}$**.Encryption Query: Same as in **Game $G_{2.1}$**.Decryption Queries: Same as in **Game $G_{2.1}$**.Authorization Query: Same as in **Game $G_{2.1}$**.
***Challenge:***$A_{2}$ chooses $M_{0},M_{1}\subset M$ randomly and sends them to S. Then, S takes b∈{0,1}, $W^{*}\in {\left\{0,1\right\}}^{l+l_{1}}$ and $r_{1},r_{2}\in {\left\{0,1\right\}}^{l_{1}}$. It then outputs $CT^{*}$ as follows:
$$C_{1}^{*}=h_{ID_{1}}^{s_{1}r_{1}}$$
$$C_{2}^{*}=M_{b}^{r_{2}}H_{2}\left(U_{1}^{r_{1}}\right)$$
$$C_{3}^{*}=g^{r_{1}}$$
$$C_{4}^{*}=g^{sr_{2}}$$
$$C_{5}^{*}=h_{ID_{1}}^{sr_{2}}$$
$$C_{6}^{*}=W^{*}$$
$$C_{7}^{*}=H_{5}(C_{1}^{*}\|C_{2}^{*}\|C_{3}^{*}\|C_{4}^{*}\|C_{5}^{*}\|C_{6}^{*}\left\|h_{ID_{1}}^{s}\right).$$
$$C_{8}^{*}=H_{4}(C_{1}^{*}\|C_{2}^{*}\|C_{3}^{*}\|C_{4}^{*}\|C_{5}^{*}\|C_{6}^{*}\|C_{7}^{*}\|M_{b}\left\|r_{1}\right)$$
where $U_{1}=e\left(h_{ID_{1}}^{s},g^{s_{1}s_{2}}\right)$. It outputs the ciphertext $CT^{*}=\left(C_{1}^{*},C_{2}^{*},C_{3}^{*},C_{4}^{*},C_{5}^{*},C_{6}^{*},C_{7}^{*},C_{8}^{*}\right)$, and adds $(e{\left(h_{ID_{2}},g^{s}\right)}^{r_{1}},W^{*}⊕\left(M^{*}\|r_{1}\right))$ into $H_{3}$.Finally, it sends $CT^{*}$ to $A_{2}$ as the challenge ciphertext.***Phase 2:***$A_{2}$ performs the same queries as in ***Phase 1***, where the constraint is that $CT^{*}$ does not appear in the decryption queries, and if $A_{2}$ asks for the decryption of $CT^{*}=\left(C_{1}^{*},C_{2}^{*},C_{3}^{*},C_{4}^{*},C_{5}^{*},C_{6}^{{}^{\prime }},C_{7}^{*},C_{8}^{*}\right)$, where $C_{6}^{{}^{\prime }}\neq C_{6}^{*}$, S outputs ⊥.***Guess:***$A_{2}$ outputs $b^{*}\in \left\{0,1\right\}$. □

Let $E_{2.2}$ be the event that $b=b^{*}$ in **Game $G_{2.2}$**.

Because $C_{6}^{{}^{\prime }}$ is a random value in **Game $G_{2.1}$** and **Game $G_{2.2}$**, the challenge ciphertexts generated in **Game $G_{2.1}$** and **Game $G_{2.2}$** follow the same distribution. Therefore, if the event $E_{2}$ does not occur, **Game $G_{2.2}$** is identical to **Game $G_{2.1}$**. And we can figure out that
$$|Pr\left[E_{2.2}\right]-Pr\left[E_{2.1}\right]|\leq Pr\left[E_{2}\right].$$

Next, we show that the probability of event $E_{2}$ occurring in **Game $G_{2.2}$** is negligible.

**Lemma** **2.***When the C-BDH problem is intractable, there is negligible probability that the event $E_{2}$ will happen in***Game**$G_{2.2}$.

**Proof.** Suppose that $Pr[E_{2}]$ is non-negligible; we can construct a simulator S to break the C-BDH assumption by using the $A_{2}$’s attacks. With the tuple $\left(e,G,G_{T},g,g^{a},g^{c},g^{d}\right)$, the aim is to obtain $e{\left(g,g\right)}^{acd}$.**Setup:**S randomly select $ID_{1},ID_{2}$ as a challenger sender and a challenger receiver, respectively. Then, S gives the public key and $ID_{1},ID_{2}$ to $A_{2}$. S runs the algorithm **Setup**($1^{l}$) to create the system parameters $sp=(g,G,G_{T},g^{s},e,H_{1},H_{2},$ $H_{3},H_{4},H_{5})$, runs the algorithm **KeyGengroup**(sp) to create a group private key $gsk=(s_{1},s_{2})$, runs the algorithm **Join**($gsk,h_{ID}$) to create the group public key $gpk=(h_{ID}^{s_{1}},g^{s_{1}},g^{s_{1}s_{2}})$ for user ID, and runs **Auth**(gsk) to create the group trapdoor $gtd=s_{2}$.**Phase 1:**S responds to the queries made by $A_{2}$ in the following ways:
$H_{1}$-query(ID), $H_{2}$-query($\theta_{i}$), $H_{5}$-query($\phi_{i}$), and $H_{4}$-query($\rho_{i}$) are the same as in **Game $G_{2.1}$**.$H_{3}$-query($\mu_{i}$) is the same as in **Game $G_{2.1}$**, except that $A_{2}$ asks for $e(C_{3},h_{ID_{2}}^{s})$.Extract Query(ID): Same as in **Game $G_{2.1}$**.Encryption Query: Same as in **Game $G_{2.1}$**, except that for the query $\left(ID_{2},*,*\right)$, S selects $r_{1},r_{2}\in {\left\{0,1\right\}}^{l_{1}}$ randomly and outputs a ciphertext $CT=(C_{1},C_{2},C_{3},C_{4},C_{5},C_{6},C_{7},C_{8})$ as follows:S performs the $H_{1}$-query($ID_{i}$) and $H_{1}$-query($ID_{j}$) to obtain $\alpha_{i}$ and $\alpha_{j}$, respectively, the $H_{2}$-query($e{\left(C_{1},g^{s}\right)}^{s_{2}}$) to obtain $\vartheta_{i}$, the $H_{3}$-query($e{\left(\alpha_{j},g^{s}\right)}^{r_{1}}$) to obtain $\nu_{i}$, the $H_{5}$-query($C_{1}\|C_{2}\|C_{3}\|C_{4}\|C_{5}\left\|C_{6}\right\|h_{ID_{i}}^{s}$) to obtain $\varphi_{i}$, and the $H_{4}$-query($C_{1}\|C_{2}\|C_{3}\|C_{4}\|C_{5}\|C_{6}\|C_{7}\left\|M\right\|r_{1}$) to obtain $\xi_{i}$.
$$C_{1}=\alpha_{i}^{s_{1}r_{1}}$$
$$C_{2}=M^{r_{2}}\vartheta_{i}$$
$$C_{3}=g^{r_{2}}$$
$$C_{4}=g^{sr_{2}}$$
$$C_{5}=\alpha_{j}^{sr_{1}}$$
$$C_{6}=\nu_{i}⊕\left(M\|r_{1}\right)$$
$$C_{7}=\varphi_{i}$$
$$C_{8}=\xi_{i}.$$S adds $\left(e{\left(C_{1},g^{s}\right)}^{s_{2}},\vartheta_{i}\right)$ to the $H_{2}$ list, adds $\left(e{\left(\alpha_{j},g^{s}\right)}^{r_{1}},\nu_{i}\right)$ to the $H_{3}$ list, adds ($C_{1}\|C_{2}\|C_{3}\|C_{4}\|C_{5}\|C_{6}\|C_{7}\left\|M\right\|r_{1},\xi_{i}$) to the $H_{4}$ list, and adds ($C_{1}\|C_{2}\|C_{3}\|C_{4}\|C_{5}\left\|C_{6}\right\|h_{ID_{i}}^{s},\varphi_{i}$) to the $H_{5}$ list.Decryption Queries: Same as in **Game $G_{2.1}$**.
***Challenge:***$A_{2}$ chooses $M_{0},M_{1}\subset M$ randomly and sends them to S. Then, S takes b∈{0,1}, $W^{*}\in {\left\{0,1\right\}}^{l+l_{1}}$, and $r_{1},r_{2}\in {\left\{0,1\right\}}^{l_{1}}$. Then, it outputs $CT^{*}$ as follows:
$$C_{1}^{*}=h_{ID_{1}}^{s_{1}r_{1}}$$
$$C_{2}^{*}=M_{b}^{r_{2}}H_{2}\left(U_{1}^{r_{1}}\right)$$
$$C_{3}^{*}=g^{r_{1}}$$
$$C_{4}^{*}=g^{sr_{2}}$$
$$C_{5}^{*}=h_{ID_{1}}^{sr_{1}}$$
$$C_{6}^{*}=W^{*}$$
$$C_{7}^{*}=H_{5}(C_{1}^{*}\|C_{2}^{*}\|C_{3}^{*}\|C_{4}^{*}\|C_{5}^{*}\|C_{6}^{*}\left\|h_{ID_{1}}^{s}\right).$$
$$C_{8}^{*}=H_{4}(C_{1}^{*}\|C_{2}^{*}\|C_{3}^{*}\|C_{4}^{*}\|C_{5}^{*}\|C_{6}^{*}\|C_{7}^{*}\|M_{b}\left\|r_{1}\right)$$
where $U_{1}=e\left(h_{ID_{1}}^{s},g^{s_{1}s_{2}}\right)$. It outputs the ciphertext $CT^{*}=\left(C_{1}^{*},C_{2}^{*},C_{3}^{*},C_{4}^{*},C_{5}^{*},C_{6}^{*},C_{7}^{*},C_{8}^{*}\right)$, and adds $(e{\left(h_{ID_{2}},g^{s}\right)}^{r_{1}},W^{*}⊕\left(M_{b}\|r_{1}\right))$ into $H_{3}$.Finally, it sends $CT^{*}$ to $A_{2}$ as the challenge ciphertext.***Phase 2:***$A_{2}$ performs the same queries as in ***Phase 1***, where the constraint is that $CT^{*}$ does not appear in the decryption queries, and if $A_{2}$ asks for the decryption of $CT^{*}=\left(C_{1}^{*},C_{2}^{*},C_{3}^{*},C_{4}^{*},C_{5}^{*},C_{6}^{{}^{\prime }},C_{7}^{*},C_{8}^{*}\right)$, where $C_{6}^{{}^{\prime }}\neq C_{6}^{*}$, S outputs ⊥.***Guess:***$A_{2}$ outputs $b^{*}\in \left\{0,1\right\}$. □

## 5. Performance Comparison

In this section, a performance comparison between the presented T-GIBEwET scheme and other related schemes is discussed. As illustrated in Table 2, our proposal supports the traceability function and others do not. In Table 3, the comparison of efficiency with PKEwET variants is shown. The second to sixth columns reveal the computational efficiency for the algorithms of encryption, decryption, authorization, testing, and tracing. Compared to [7,16,17,35], the proposed T-GIBEwET scheme is more efficient than [7,16,17] in the decryption algorithm and more efficient than [17] in the authorization algorithm. Both authorization and tracking are supported in this paper.

## 6. Conclusions

In this paper we analyzed the PKEwET scheme, pointed out that the PKEwET algorithm is unable to keep track of ciphertexts in the cloud sever, and proposed the a traceable group ID-based encryption with an equality test scheme (T-GIBEwET). The T-GIBEwET algorithm is endowed with a special function: the users who are authorized by a trapdoor can test the ciphertexts in the cloud sever. Moreover, the proposed scheme supports the traceability function.

To simplify the public key management mechanism, the proposed scheme was designed with ID-based encryption. According to the competence of different users, the proposal can resist OW-CCA and IND-CCA security. Additionally, the T-GIBEwET scheme can resist a plaintext space attack.

Compared with other existing works, our proposal is more practical for use in cloud computing services.

## Figures and Tables

**Figure 1 entropy-24-00309-f001:**
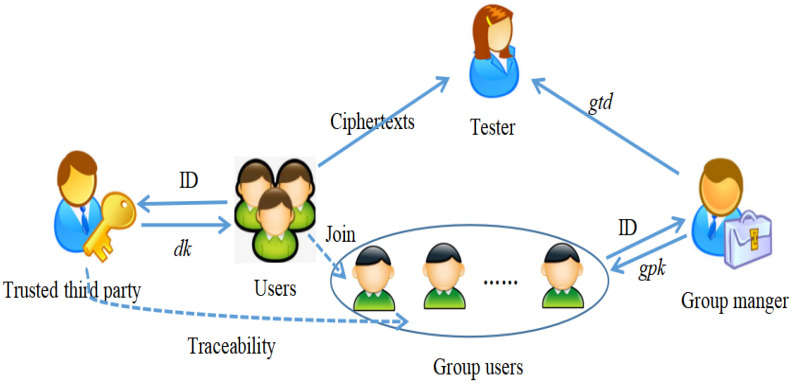
System Model.

**Figure 2 entropy-24-00309-f002:**
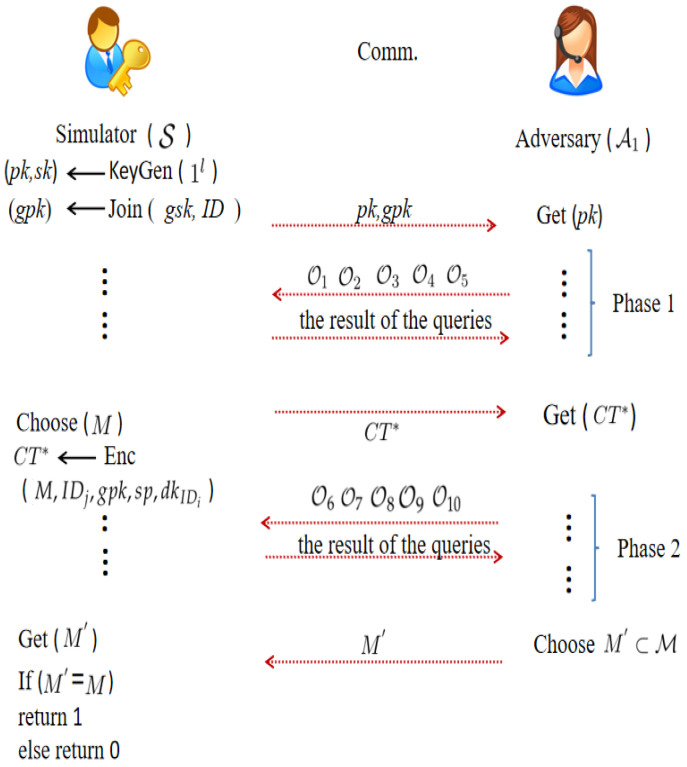
OW-CCA security model.

**Figure 3 entropy-24-00309-f003:**
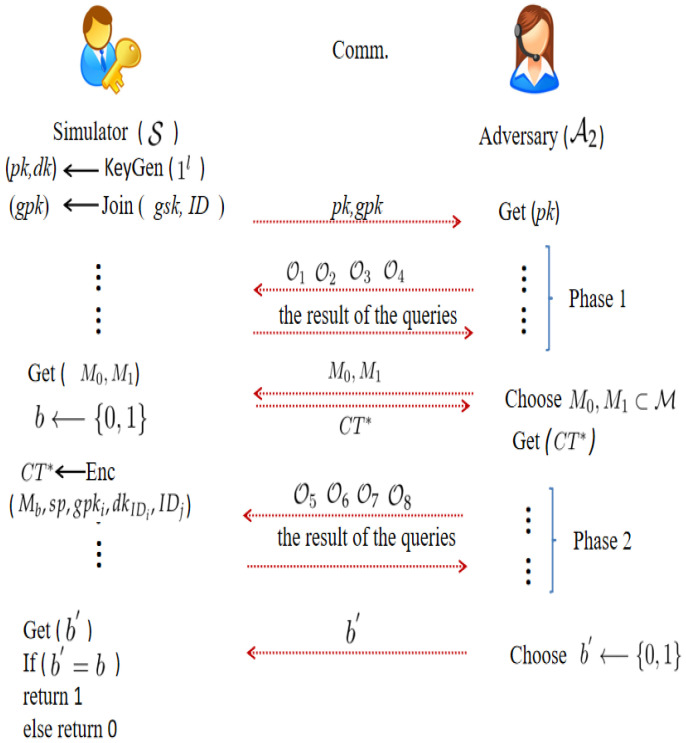
IND-CCA Security Model.

**Table 1 entropy-24-00309-t001:** Symbols used in the proposed scheme.

Symbol	Description
*l*	A security parameter
*G*	A cyclic group
*g*	The generator of G
*M*	The plaintext
CT	The ciphertext
$CT^{*}$	The challenge ciphertext
M	The message space
*Z*	Set of integers
*H*	A hash function
*s*	The master key (keep it as a secret)
ID	A user’s identity
gsk	The group secret key (kept as a secret by group manager)
gpk	The group public key (share to all users in the group)
$dk_{ID}$	A user’s secret key (keep it as a secret)
A	The adversary
S	The simulator

**Table 2 entropy-24-00309-t002:** Comparison with other schemes.

Scheme	Authorized	Ciphertext Test	Traceable
[7]	-	*√*	-
[16]	*√*	*√*	-
[17]	*√*	*√*	-
[35]	*√*	*√*	-
T-GIBEwET	*√*	*√*	*√*

**Table 3 entropy-24-00309-t003:** Comparison of efficiency with other schemes.

Scheme	$C_{\mathrm{Enc}}$	$C_{\mathrm{Dec}}$	$C_{\mathrm{Auth}}$	$C_{\mathrm{Test}}$	$C_{\mathrm{Trac}}$
[7]	3E	3E	-	2P	-
[16]	5E+2P	2E+2P	0	4P	-
[17]	(n+2)E+2P	(n+1)P+E	nE	nP	-
[35]	5E	2E	0	2E+2P	-
[34]	4E	2E	0	2E	-
T-GIBEwET	7E+2P	E+P	0	2E+4P	2P

## Data Availability

Not applicable.

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
