# Peer review of "Traceable Scheme of Public Key Encryption with Equality Test"

_entropy, 2022, doi:10.3390/e24030309_

Round 1

Reviewer 1 Report

The authors have improved the article significantly. 

Abstract is fine. However, the last sentence of the abstract can be removed. 

In introduction, the problem statement and research gap are very much clear. Subsequently, the contributions are linked with the identified research gap. 

Section 2 has summarized the necessary background knowledge in an easy to understand way. 

The proposed constructions and security analysis are logical. Furthermore, the significance of the proposal is illustrated through performance comparison. 

To summarize, the article has been improved and can be published. 

Author Response

Thank you very much for your kindly comments of our article.

Reviewer 2 Report

The authors have addressed all issues raised in my previuos review reports. I think now the paper can be published in this journal.

<Minor comment>

- At page 4, remove "=s_2" from "gtd=s_2" at the end of the description of the Auth algorithm.

Author Response

(The authors gave the same response as above.)

Reviewer 3 Report

The authors have solved my questions and revised the manuscript according to the comments.

Author Response

Thank you very much for your kindly comments of our article.

This manuscript is a resubmission of an earlier submission. The following is a list of the peer review reports and author responses from that submission.

Round 1

Reviewer 1 Report

The paper is intending to introduce traceability in public key encryption supporting equality test (PKEwET).

The following major limitations have been observed:

(1) Problem statement: In introduction, the problem is not defined at all. It is therefore required to first provide a paragraph on PKEwET. The subsequent paragraph should provide some information about traceability and its importance in PKEwET.

(2) Research Gape: Section 1.1 provides a good summary of Equality Test Schemes in general. However, the article is about PKEwET. Therefore, it is better to provide a particular focus on PKEwET. What are the limitations of existing works on PKEwET ? How a traceable scheme can really solve the identified limitations ?

(3) Contributions: The first line in Section 1.2 states that “In this paper, the ability of traceable is introduced into group ID-based encryption (GIBE) scheme”. Here the reader is surprised to find a new term GIBE, as no information is provided on GIBE in preceding paragraphs. Authors further state that “The motivation is to make GIBE supporting traceability and equality test function to the ciphertexts”. This motivation was not mentioned in problem statement part of the article. To summarize, the contribution is not linked with the problem statement and research gap.

(4) Proposed work: Before stating the contributions in Introduction section, it is critical to state the major steps in proposed work. An illustrative diagram van be helpful. Based on major steps, the contributions (from scientific or engineering point of view) can be deduced.

(5) Validation, outcomes and significance: In addition to the major steps in proposed work, the validation, outcomes of the proposed approach as well as their significance must be stated in Introduction of the article. Without the validation information, it is not confirmed that the article is technically viable or not ? Without the description of significance, the contributions of article towards the body of knowledge can not be judged.  

(6) Presentation and Technical contents in Section 2 and Section 3: The presentation of contents in these sections is below average and difficult to understand. Why the presented information is necessary to include ? What’s the reason for making two different sections ?? The information in  Section 2 and Section 3 can be merged into a single section.

(7) Novelty in Section 4: The 8 steps in Section 4 are typical. What’s the contributions or novelty as compared to state of the art ??

(8) Security analysis: The contents are presented without an overall analysis approach. It is essential to first describe the analysis approach, the tools used. What are the parameters for analysis ?? The analysis results should be visualized through some illustrative figures.

Minor comments:

The article is required to be comprehensively reviewed from English Language point of view. Some issues are highlighted as an example:

L 15: The following sentence required to be rephrased: “However, the encrypted data brings much trouble for using effectively information, unable to search on encrypted data.”

L 54: The following sentence required to be rephrased:  “In this paper, the ability of traceable is introduced into group ID-based encryption (GIBE) scheme.”

L:444 and proposes the the traceable group …. (delete repeated words)

Author Response

Thank you sincerely for your comments on our work.

Reviewer 2 Report

This paper proposes a traceable group identity-based encryption with equality test scheme (T-GIBEwET). Unfortunately, however, there are several unclear parts to be published.

  1. What is the T-GIBEwET? Throughout the paper, the authors did not provide the formal/information definitions for T-GIBEwET. So, I cannot confirm the correctness of the proposed scheme at all. The authors first need to elaborate the model of T-GIBEwET and then provide formal definitions of the scheme including its correctness that they want to propose.

  1. Though the authors provide two security definitions for T-GIBEwET, they are also informal. By following existing works on public key/identity-based encryption with equality test, the authors need to provide more formal security definitions for T-GIBEwET. For example, the descriptions of oracles O_1, O_2, O_3, O_4 and O_5 should be more detailed. In the IND-CCA security definition, there is no restriction on authorization queries in the current version. But, the adversary in the IND-CCA security game should not be allowed to query on the target identity on authorization queries because the adversary in the IND-CCA security game cannot have a trapdoor.

  1. The description of the proposed scheme is unclear. The encryption algorithm takes pk and sk as inputs. What are pk and sk? In the encryption algorithm, it requires g^s to compute U_2. But, g^s is not published by other algorithms. How does the encryption algorithm get g^s? (s is the master secret key.) Who is ID_j in the first input of e to compute U_2? Is it a receiver?
    In summary, the description is not clear, so it is hard for me to understand the scheme and its correctness.

  1. In the introduction, the authors enumerate four items as key contributions of this paper. Among them, the third one is about plaintext space attack. I read the paper carefully, but I cannot find any part about plaintext space attack. To provide it as one of key contributions, the authors need to clearly elaborate it in the main body.
    (If the plaintext space attack means a type of attacks that the adversary generates a ciphertext by selecting and encrypting a message and then comparing it with the target ciphertext, it is well-known that PKEwET cannot avoid this attack. So, it is generally assumed that the size of plaintext space is exponential in the security parameter and the min-entropy of the message distribution is sufficiently higher than the security parameter.)

  2. The authors claimed that the proposed scheme is more efficient and more practical in many scenarios. However, unfortunately, I cannot agree with this claim. Of course, (if the proposed scheme is correct and secure), it is the first scheme that supports in the area of public key/identity-based encryption with equality test. So, it supports additional functionality, compared to the existing schemes. However, the proposed scheme is not more efficient than others. For example, the instantiation of PKEET in [LLSW19] requires 4 exponentiations, 2 exponentiations and 2 exponentiations for encryption, decryption, and test respectively, while the proposed scheme in this paper requires 7 exponentiations and 2 pairings, 1 exponentiation and 1 pairing, and 2 exponentiations and 4 pairings for them, respectively. (Generally, pairing operations are much expensive than exponentiation operations.) Thus, it is overstated that the proposed scheme is more efficient at least.

From the above reasons, I cannot support the acceptance of this paper.

[LLSW19] H.T. Lee, S. Ling, J. H. Seo, and H. Wang, Public Key Encryption with Equality Test from Generic Assumptions in the Random Oracle Model, Information Sciences, Vol. 500, pp. 15-33, 2019.

Author Response

(The authors gave the same response as above.)

Reviewer 3 Report

This manuscript entitled "Traceable scheme of public key encryption with equality test" tends to introduce the traceability of ciphertexts into encryption with equality test primitive. Based on a GIBE scheme, this manuscript adds the equality test and traceable functions. This idea is innovative, while there are several problems that need to be revised carefully.

1. Undefined "traceability". The "traceability" is not clearly defined in the Abstract and Contribution parts, which confuses me on which party can be traceable at the beginning until "our construction" part. Authors should revise the manuscript to stress out the traceability here refers to the ciphertexts.

2. Informal Figure.  Authors are suggested to revise or delete Fig. 1, it seems like a draft, instead of a formal figure in an academic paper.

3. Inaccurate definition. In Sec. 2.1, the authors define OW-CCA and IND security in an informal way. This is not necessary since the security models are defined in Sec. 3. While, in the Security Model part (Fig 2 and 3), authors should know it is adversary, not simulator, in the security model definitions.

4. Lack of explanation for symbols in the scheme. According to the scheme's construction, for each user with ID, there are one secret key dkID and a group public key gpk? Should not the group public key be a public and shared public key for all the users in the group?

5. In the tracing algorithm, the algorithm checks whether D1=D2 to determine whether the ciphertext is generated by the user with ID. Two problems here. First, s1 can be omitted in D1 and D2. Next, it seems the algorithm does not work. For example, one can simply change C_4, C5 of a ciphertext to C4'=g^{s' r'}, C_5=h_{ID'}^{s', r'} with random chosen s', r', which enables the equation D1=D2 holds for h_{ID'}.

6. In the testing algorithm, to compute M_i^r, g^s is utilized. However, g^s is not a public parameter in the construction. 

In conclusion, I think the authors should revise this manuscript to solve the above problems first.

Author Response

(The authors gave the same response as above.)
